# Grammar as a Foreign Language

**Oriol Vinyals**[*]
Google
vinyals@google.com

**Lukasz Kaiser**[*]
Google
lukaszkaiser@google.com

**Terry Koo**
Google
terrykoo@google.com

**Slav Petrov**
Google
slav@google.com

**Ilya Sutskever**
Google
ilyasu@google.com

**Geoffrey Hinton**
Google
geoffhinton@google.com

## Abstract

Syntactic constituency parsing is a fundamental problem in natural language processing and has been the subject of intensive research and engineering for decades. As a result, the most accurate parsers are domain specific, complex, and inefficient. In this paper we show that the domain agnostic attention-enhanced sequence-to-sequence model achieves state-of-the-art results on the most widely used syntactic constituency parsing dataset, when trained on a large synthetic corpus that was annotated using existing parsers. It also matches the performance of standard parsers when trained only on a small human-annotated dataset, which shows that this model is highly data-efficient, in contrast to sequence-to-sequence models without the attention mechanism. Our parser is also fast, processing over a hundred sentences per second with an unoptimized CPU implementation.

## 1  Introduction

Syntactic constituency parsing is a fundamental problem in linguistics and natural language processing that has a wide range of applications. This problem has been the subject of intense research for decades, and as a result, there exist highly accurate domain-specific parsers. The computational requirements of traditional parsers are cubic in sentence length, and while linear-time shift-reduce constituency parsers improved in accuracy in recent years, they never matched state-of-the-art. Furthermore, standard parsers have been designed with parsing in mind; the concept of a parse tree is deeply ingrained into these systems, which makes these methods inapplicable to other problems.

Recently, Sutskever et al. [1] introduced a neural network model for solving the general sequence-to-sequence problem, and Bahdanau et al. [2] proposed a related model with an attention mechanism that makes it capable of handling long sequences well. Both models achieve state-of-the-art results on large scale machine translation tasks (e.g., [3, 4]). Syntactic constituency parsing can be formulated as a sequence-to-sequence problem if we linearize the parse tree (cf. Figure 2), so we can apply these models to parsing as well.

Our early experiments focused on the sequence-to-sequence model of Sutskever et al. [1]. We found this model to work poorly when we trained it on standard human-annotated parsing datasets (1M tokens), so we constructed an artificial dataset by labelling a large corpus with the BerkeleyParser.

---

[*]Equal contribution

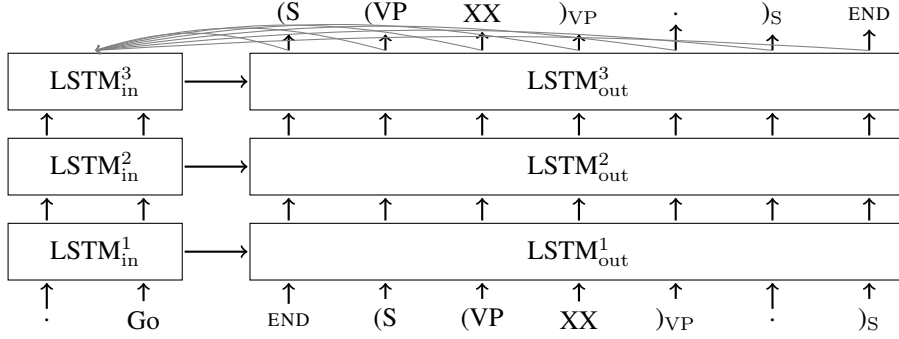

Figure 1: A schematic outline of a run of our LSTM+A model on the sentence "Go.". See text for details.

To our surprise, the sequence-to-sequence model matched the BerkeleyParser that produced the annotation, having achieved an F1 score of 90.5 on the test set (section 23 of the WSJ).

We suspected that the attention model of Bahdanau et al. [2] might be more data efficient and we found that it is indeed the case. We trained a sequence-to-sequence model with attention on the small human-annotated parsing dataset and were able to achieve an F1 score of 88.3 on section 23 of the WSJ without the use of an ensemble and 90.5 with an ensemble, which matches the performance of the BerkeleyParser (90.4) when trained on the same data.

Finally, we constructed a second artificial dataset consisting of only high-confidence parse trees, as measured by the agreement of two parsers. We trained a sequence-to-sequence model with attention on this data and achieved an F1 score of 92.1 on section 23 of the WSJ, matching the state-of-the-art. This result did not require an ensemble, and as a result, the parser is also very fast.

## 2   LSTM+A Parsing Model

Let us first recall the sequence-to-sequence LSTM model. The Long Short-Term Memory model of [5] is defined as follows. Let $x_t$, $h_t$, and $m_t$ be the input, control state, and memory state at timestep $t$. Given a sequence of inputs $(x_1, \ldots, x_T)$, the LSTM computes the $h$-sequence $(h_1, \ldots, h_T)$ and the $m$-sequence $(m_1, \ldots, m_T)$ as follows.

$$
\begin{aligned}
i_t &= \text{sigm}(W_1 x_t + W_2 h_{t-1}) \\
i'_t &= \tanh(W_3 x_t + W_4 h_{t-1}) \\
f_t &= \text{sigm}(W_5 x_t + W_6 h_{t-1}) \\
o_t &= \text{sigm}(W_7 x_t + W_8 h_{t-1}) \\
m_t &= m_{t-1} \odot f_t + i_t \odot i'_t \\
h_t &= m_t \odot o_t
\end{aligned}
$$

The operator $\odot$ denotes element-wise multiplication, the matrices $W_1, \ldots, W_8$ and the vector $h_0$ are the parameters of the model, and all the nonlinearities are computed element-wise.

In a deep LSTM, each subsequent layer uses the $h$-sequence of the previous layer for its input sequence $x$. The deep LSTM defines a distribution over output sequences given an input sequence:

$$
\begin{aligned}
P(B|A) &= \prod_{t=1}^{T_B} P(B_t | A_1, \ldots, A_{T_A}, B_1, \ldots, B_{t-1}) \\
&\equiv \prod_{t=1}^{T_B} \text{softmax}(W_o \cdot h_{T_A+t})^\top \delta_{B_t},
\end{aligned}
$$

The above equation assumes a deep LSTM whose input sequence is $x = (A_1, \ldots, A_{T_A}, B_1, \ldots, B_{T_B})$, so $h_t$ denotes $t$-th element of the $h$-sequence of topmost LSTM. The matrix $W_o$ consists of the vector representations of each output symbol and the symbol $\delta_b$

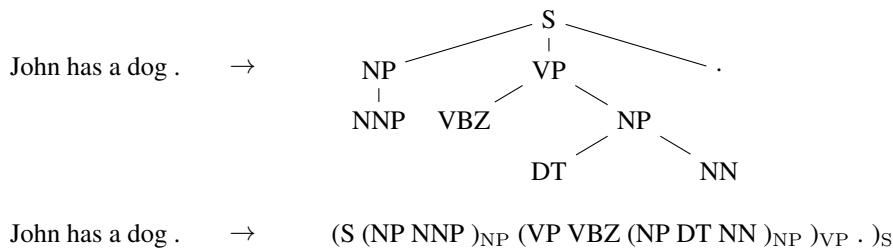

John has a dog .   →   (S (NP NNP )$_{\text{NP}}$ (VP VBZ (NP DT NN )$_{\text{NP}}$ )$_{\text{VP}}$ . )$_{\text{S}}$

Figure 2: Example parsing task and its linearization.

is a Kronecker delta with a dimension for each output symbol, so $\text{softmax}(W_o \cdot h_{T_A+t})^\top \delta_{B_t}$ is precisely the $B_t$'th element of the distribution defined by the softmax. Every output sequence terminates with a special end-of-sequence token which is necessary in order to define a distribution over sequences of variable lengths. We use two different sets of LSTM parameters, one for the input sequence and one for the output sequence, as shown in Figure 1. Stochastic gradient descent is used to maximize the training objective which is the average over the training set of the log probability of the correct output sequence given the input sequence.

## 2.1 Attention Mechanism

An important extension of the sequence-to-sequence model is by adding an attention mechanism. We adapted the attention model from [2] which, to produce each output symbol $B_t$, uses an attention mechanism over the encoder LSTM states. Similar to our sequence-to-sequence model described in the previous section, we use two separate LSTMs (one to encode the sequence of input words $A_i$, and another one to produce or decode the output symbols $B_i$). Recall that the encoder hidden states are denoted $(h_1, \ldots, h_{T_A})$ and we denote the hidden states of the decoder by $(d_1, \ldots, d_{T_B}) := (h_{T_A+1}, \ldots, h_{T_A+T_B})$.

To compute the attention vector at each output time $t$ over the input words $(1, \ldots, T_A)$ we define:

$$
\begin{aligned}
u_i^t &= v^T \tanh(W_1' h_i + W_2' d_t) \\
a_i^t &= \text{softmax}(u_i^t) \\
d_t' &= \sum_{i=1}^{T_A} a_i^t h_i
\end{aligned}
$$

The vector $v$ and matrices $W_1', W_2'$ are learnable parameters of the model. The vector $u^t$ has length $T_A$ and its $i$-th item contains a score of how much attention should be put on the $i$-th hidden encoder state $h_i$. These scores are normalized by softmax to create the attention mask $a^t$ over encoder hidden states. In all our experiments, we use the same hidden dimensionality (256) at the encoder and the decoder, so $v$ is a vector and $W_1'$ and $W_2'$ are square matrices. Lastly, we concatenate $d_t'$ with $d_t$, which becomes the new hidden state from which we make predictions, and which is fed to the next time step in our recurrent model.

In Section 4 we provide an analysis of what the attention mechanism learned, and we visualize the normalized attention vector $a^t$ for all $t$ in Figure 4.

## 2.2 Linearizing Parsing Trees

To apply the model described above to parsing, we need to design an invertible way of converting the parse tree into a sequence (linearization). We do this in a very simple way following a depth-first traversal order, as depicted in Figure 2.

We use the above model for parsing in the following way. First, the network consumes the sentence in a left-to-right sweep, creating vectors in memory. Then, it outputs the linearized parse tree using information in these vectors. As described below, we use 3 LSTM layers, reverse the input sentence and normalize part-of-speech tags. An example run of our LSTM+A model on the sentence "Go." is depicted in Figure 1 (top gray edges illustrate attention).

### 2.3 Parameters and Initialization

**Sizes.**   In our experiments we used a model with 3 LSTM layers and 256 units in each layer, which we call LSTM+A. Our input vocabulary size was 90K and we output 128 symbols.

**Dropout.**   Training on a small dataset we additionally used 2 dropout layers, one between LSTM[1] and LSTM[2], and one between LSTM[2] and LSTM[3]. We call this model LSTM+A+D.

**POS-tag normalization.**   Since part-of-speech (POS) tags are not evaluated in the syntactic parsing F1 score, we replaced all of them by "XX" in the training data. This improved our F1 score by about 1 point, which is surprising: For standard parsers, including POS tags in training data helps significantly. All experiments reported below are performed with normalized POS tags.

**Input reversing.**   We also found it useful to reverse the input sentences but not their parse trees, similarly to [1]. Not reversing the input had a small negative impact on the F1 score on our development set (about 0.2 absolute). All experiments reported below are performed with input reversing.

**Pre-training word vectors.**   The embedding layer for our 90K vocabulary can be initialized randomly or using pre-trained word-vector embeddings. We pre-trained skip-gram embeddings of size 512 using `word2vec` [6] on a 10B-word corpus. These embeddings were used to initialize our network but not fixed, they were later modified during training. We discuss the impact of pre-training in the experimental section.

We do not apply any other special preprocessing to the data. In particular, we do not binarize the parse trees or handle unaries in any specific way. We also treat unknown words in a naive way: we map all words beyond our 90K vocabulary to a single UNK token. This potentially underestimates our final results, but keeps our framework task-independent.

## 3 Experiments

### 3.1 Training Data

We trained the model described above on 2 different datasets. For one, we trained on the standard WSJ training dataset. This is a very small training set by neural network standards, as it contains only 40K sentences (compared to 60K examples even in MNIST). Still, even training on this set, we managed to get results that match those obtained by domain-specific parsers.

To match state-of-the-art, we created another, larger training set of $\sim$11M parsed sentences (250M tokens). First, we collected all publicly available treebanks. We used the OntoNotes corpus version 5 [7], the English Web Treebank [8] and the updated and corrected Question Treebank [9].[1] Note that the popular Wall Street Journal section of the Penn Treebank [10] is part of the OntoNotes corpus. In total, these corpora give us $\sim$90K training sentences (we held out certain sections for evaluation, as described below).

In addition to this gold standard data, we use a corpus parsed with existing parsers using the "tri-training" approach of [11]. In this approach, two parsers, our reimplementation of Berkeley-Parser [12] and a reimplementation of ZPar [13], are used to process unlabeled sentences sampled from news appearing on the web. We select only sentences for which both parsers produced the same parse tree and re-sample to match the distribution of sentence lengths of the WSJ training corpus. Re-sampling is useful because parsers agree much more often on short sentences. We call the set of $\sim$11 million sentences selected in this way, together with the $\sim$90K golden sentences described above, the *high-confidence corpus*.

After creating this corpus, we made sure that no sentence from the development or test set appears in the corpus, also after replacing rare words with "unknown" tokens. This operation guarantees that we never see any test sentence during training, but it also lowers our F1 score by about 0.5 points. We are not sure if such strict de-duplication was performed in previous works, but even with this, we still match state-of-the-art.

| Parser | Training Set | WSJ 22 | WSJ 23 |
|---|---|---|---|
| baseline LSTM+D | WSJ only | < 70 | < 70 |
| LSTM+A+D | WSJ only | 88.7 | 88.3 |
| LSTM+A+D ensemble | WSJ only | 90.7 | 90.5 |
| baseline LSTM | BerkeleyParser corpus | 91.0 | 90.5 |
| LSTM+A | high-confidence corpus | **92.8** | **92.1** |
| Petrov et al. (2006) [12] | WSJ only | 91.1 | 90.4 |
| Zhu et al. (2013) [13] | WSJ only | N/A | 90.4 |
| Petrov et al. (2010) ensemble [14] | WSJ only | 92.5 | 91.8 |
| Zhu et al. (2013) [13] | semi-supervised | N/A | 91.3 |
| Huang & Harper (2009) [15] | semi-supervised | N/A | 91.3 |
| McClosky et al. (2006) [16] | semi-supervised | 92.4 | **92.1** |

Table 1: F1 scores of various parsers on the development and test set. See text for discussion.

In earlier experiments, we only used one parser, our reimplementation of BerkeleyParser, to create a corpus of parsed sentences. In that case we just parsed ∼7 million senteces from news appearing on the web and combined these parsed sentences with the ∼90K golden corpus described above. We call this the *BerkeleyParser corpus*.

## 3.2 Evaluation

We use the standard EVALB tool[2] for evaluation and report F1 scores on our developments set (section 22 of the Penn Treebank) and the final test set (section 23) in Table 1.

First, let us remark that our training setup differs from those reported in previous works. To the best of our knowledge, no standard parsers have ever been trained on datasets numbering in the hundreds of millions of tokens, and it would be hard to do due to efficiency problems. We therefore cite the semi-supervised results, which are analogous in spirit but use less data.

Table 1 shows performance of our models on the top and results from other papers at the bottom. We compare to variants of the BerkeleyParser that use self-training on unlabeled data [15], or built an ensemble of multiple parsers [14], or combine both techniques. We also include the best linear-time parser in the literature, the transition-based parser of [13].

It can be seen that, when training on WSJ only, a baseline LSTM does not achieve any reasonable score, even with dropout and early stopping. But a single attention model gets to 88.3 and an ensemble of 5 LSTM+A+D models achieves 90.5 matching a single-model BerkeleyParser on WSJ 23. When trained on the large high-confidence corpus, a single LSTM+A model achieves 92.1 and so matches the best previous single model result.

**Generating well-formed trees.** The LSTM+A model trained on WSJ dataset only produced malformed trees for 25 of the 1700 sentences in our development set (1.5% of all cases), and the model trained on full high-confidence dataset did this for 14 sentences (0.8%). In these few cases where LSTM+A outputs a malformed tree, we simply add brackets to either the beginning or the end of the tree in order to make it balanced. It is worth noting that all 14 cases where LSTM+A produced unbalanced trees were sentences or sentence fragments that did not end with proper punctuation. There were very few such sentences in the training data, so it is not a surprise that our model cannot deal with them very well.

**Score by sentence length.** An important concern with the sequence-to-sequence LSTM was that it may not be able to handle long sentences well. We determine the extent of this problem by partitioning the development set by length, and evaluating BerkeleyParser, a baseline LSTM model without attention, and LSTM+A on sentences of each length. The results, presented in Figure 3, are surprising. The difference between the F1 score on sentences of length upto 30 and that upto 70 is 1.3 for the BerkeleyParser, 1.7 for the baseline LSTM, and 0.7 for LSTM+A. So already the baseline LSTM has similar performance to the BerkeleyParser, it degrades with length only slightly. Surprisingly, LSTM+A shows *less* degradation with length than BerkeleyParser – a full $O(n^3)$ chart parser that uses a lot more memory.

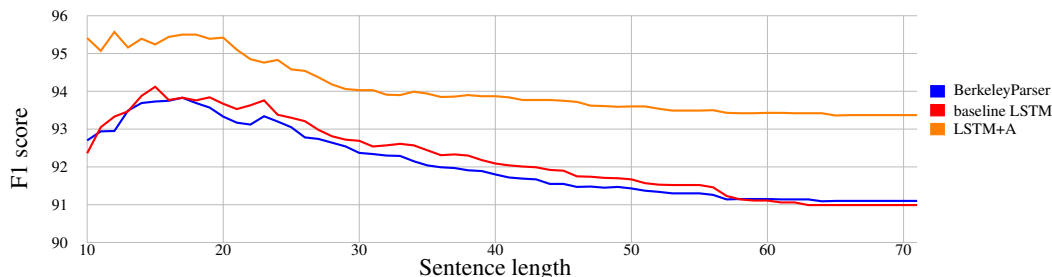

Figure 3: Effect of sentence length on the F1 score on WSJ 22.

**Beam size influence.** Our decoder uses a beam of a fixed size to calculate the output sequence of labels. We experimented with different settings for the beam size. It turns out that it is almost irrelevant. We report report results that use beam size 10, but using beam size 2 only lowers the F1 score of LSTM+A on the development set by 0.2, and using beam size 1 lowers it by 0.5. Beam sizes above 10 do not give any additional improvements.

**Dropout influence.** We only used dropout when training on the small WSJ dataset and its influence was significant. A single LSTM+A model only achieved an F1 score of 86.5 on our development set, that is over 2 points lower than the 88.7 of a LSTM+A+D model.

**Pre-training influence.** As described in the previous section, we initialized the word-vector embedding with pre-trained word vectors obtained from `word2vec`. To test the influence of this initialization, we trained a LSTM+A model on the high-confidence corpus, and a LSTM+A+D model on the WSJ corpus, starting with randomly initialized word-vector embeddings. The F1 score on our development set was 0.4 lower for the LSTM+A model and 0.3 lower for the LSTM+A+D model (88.4 vs 88.7). So the effect of pre-training is consistent but small.

**Performance on other datasets.** The WSJ evaluation set has been in use for 20 years and is commonly used to compare syntactic parsers. But it is not representative for text encountered on the web [8]. Even though our model was trained on a news corpus, we wanted to check how well it generalizes to other forms of text. To this end, we evaluated it on two additional datasets:

QTB 1000 held-out sentences from the Question Treebank [9];

WEB the first half of each domain from the English Web Treebank [8] (8310 sentences).

LSTM+A trained on the high-confidence corpus (which only includes text from news) achieved an F1 score of 95.7 on QTB and 84.6 on WEB. Our score on WEB is higher both than the best score reported in [8] (83.5) and the best score we achieved with an in-house reimplementation of BerkeleyParser trained on human-annotated data (84.4). We managed to achieve a slightly higher score (84.8) with the in-house BerkeleyParser trained on a large corpus. On QTB, the 95.7 score of LSTM+A is also lower than the best score of our in-house BerkeleyParser (96.2). Still, taking into account that there were only few questions in the training data, these scores show that LSTM+A managed to generalize well beyond the news language it was trained on.

**Parsing speed.** Our LSTM+A model, running on a multi-core CPU using batches of 128 sentences on a generic unoptimized decoder, can parse over 120 sentences from WSJ per second for sentences of all lengths (using beam-size 1). This is better than the speed reported for this batch size in Figure 4 of [17] at 100 sentences per second, even though they run on a GPU and only on sentences of under 40 words. Note that they achieve 89.7 F1 score on this subset of sentences of section 22, while our model at beam-size 1 achieves a score of 93.2 on this subset.

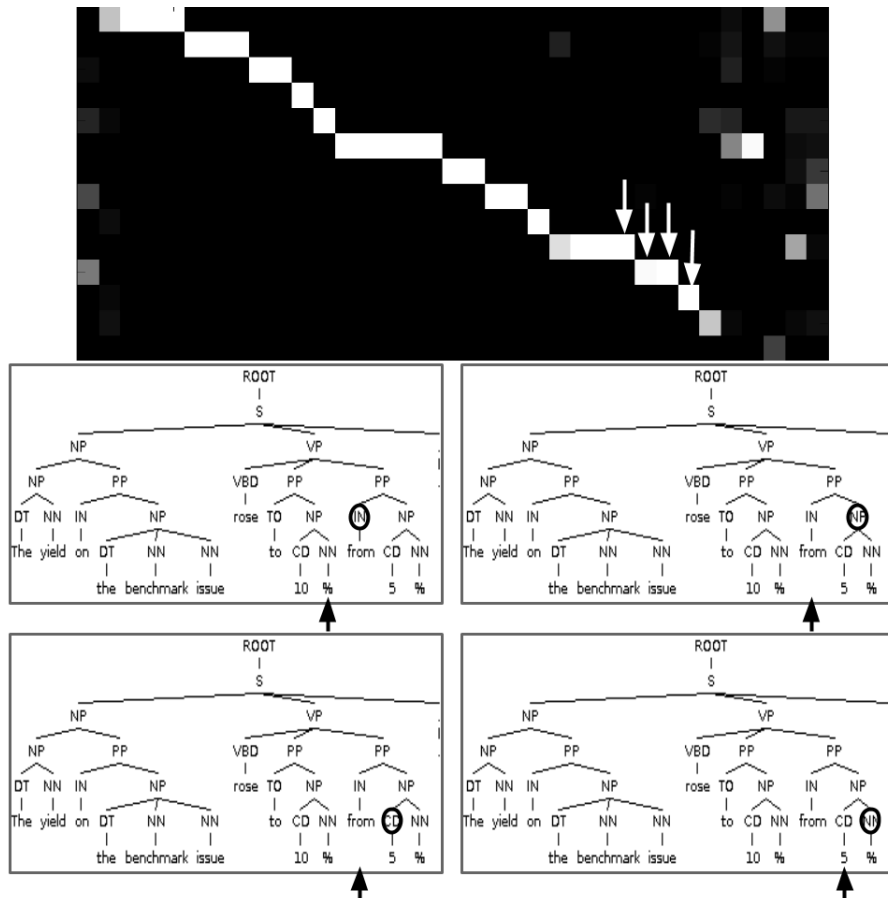

Figure 4: Attention matrix. Shown on top is the attention matrix where each column is the attention vector over the inputs. On the bottom, we show outputs for four consecutive time steps, where the attention mask moves to the right. As can be seen, every time a terminal node is consumed, the attention pointer moves to the right.

## 4 Analysis

As shown in this paper, the attention mechanism was a key component especially when learning from a relatively small dataset. We found that the model did not overfit and learned the parsing function from scratch much faster, which resulted in a model which generalized much better than the plain LSTM without attention.

One of the most interesting aspects of attention is that it allows us to visualize to interpret what the model has learned from the data. For example, in [2] it is shown that for translation, attention learns an alignment function, which certainly should help translating from English to French.

Figure 4 shows an example of the attention model trained only on the WSJ dataset. From the attention matrix, where each column is the attention vector over the inputs, it is clear that the model focuses quite sharply on one word as it produces the parse tree. It is also clear that the focus moves from the first word to the last monotonically, and steps to the right deterministically when a word is consumed.

On the bottom of Figure 4 we see where the model attends (black arrow), and the current output being decoded in the tree (black circle). This stack procedure is learned from data (as all the parameters are randomly initialized), but is not quite a simple stack decoding. Indeed, at the input side, if the model focuses on position $i$, that state has information for all words after $i$ (since we also reverse the inputs). It is worth noting that, in some examples (not shown here), the model does skip words.

# 5 Related Work

The task of syntactic constituency parsing has received a tremendous amount of attention in the last 20 years. Traditional approaches to constituency parsing rely on probabilistic context-free grammars (CFGs). The focus in these approaches is on devising appropriate smoothing techniques for highly lexicalized and thus rare events [18] or carefully crafting the model structure [19]. [12] partially alleviate the heavy reliance on manual modeling of linguistic structure by using latent variables to learn a more articulated model. However, their model still depends on a CFG backbone and is thereby potentially restricted in its capacity.

Early neural network approaches to parsing, for example by [20, 21] also relied on strong linguistic insights. [22] introduced Incremental Sigmoid Belief Networks for syntactic parsing. By constructing the model structure incrementally, they are able to avoid making strong independence assumptions but inference becomes intractable. To avoid complex inference methods, [23] propose a recurrent neural network where parse trees are decomposed into a stack of independent levels. Unfortunately, this decomposition breaks for long sentences and their accuracy on longer sentences falls quite significantly behind the state-of-the-art. [24] used a tree-structured neural network to score candidate parse trees. Their model however relies again on the CFG assumption and furthermore can only be used to score candidate trees rather than for full inference.

Our LSTM model significantly differs from all these models, as it makes no assumptions about the task. As a sequence-to-sequence prediction model it is somewhat related to the incremental parsing models, pioneered by [25] and extended by [26]. Such linear time parsers however typically need some task-specific constraints and might build up the parse in multiple passes. Relatedly, [13] present excellent parsing results with a single left-to-right pass, but require a stack to explicitly delay making decisions and a parsing-specific transition strategy in order to achieve good parsing accuracies. The LSTM in contrast uses its short term memory to model the complex underlying structure that connects the input-output pairs.

Recently, researchers have developed a number of neural network models that can be applied to general sequence-to-sequence problems. [27] was the first to propose a differentiable attention mechanism for the general problem of handwritten text synthesis, although his approach assumed a monotonic alignment between the input and output sequences. Later, [2] introduced a more general attention model that does not assume a monotonic alignment, and applied it to machine translation, and [28] applied the same model to speech recognition. [29] used a convolutional neural network to encode a variable-sized input sentence into a vector of a fixed dimension and used a RNN to produce the output sentence. Essentially the same model has been used by [30] to successfully learn to generate image captions. Finally, already in 1990 [31] experimented with applying recurrent neural networks to the problem of syntactic parsing.

# 6 Conclusions

In this work, we have shown that generic sequence-to-sequence approaches can achieve excellent results on syntactic constituency parsing with relatively little effort or tuning. In addition, while we found the model of Sutskever et al. [1] to not be particularly data efficient, the attention model of Bahdanau et al. [2] was found to be highly data efficient, as it has matched the performance of the BerkeleyParser when trained on a small human-annotated parsing dataset. Finally, we showed that synthetic datasets with imperfect labels can be highly useful, as our models have substantially outperformed the models that have been used to create their training data. We suspect it is the case due to the different natures of the teacher model and the student model: the student model has likely viewed the teacher's errors as noise which it has been able to ignore. This approach was so successful that we obtained a new state-of-the-art result in syntactic constituency parsing with a single attention model, which also means that the model is exceedingly fast. This work shows that domain independent models with excellent learning algorithms can match and even outperform domain specific models.

**Acknowledgement.** We would like to thank Amin Ahmad, Dan Bikel and Jonni Kanerva.

## Footnotes

[1] All treebanks are available through the Linguistic Data Consortium (LDC): OntoNotes (LDC2013T19), English Web Treebank (LDC2012T13) and Question Treebank (LDC2012R121).

[2]http://nlp.cs.nyu.edu/evalb/

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
