[Reviews · NeurIPS 2015]

Submitted by Assigned_Reviewer_1

The authors investigate a way of parsing natural language sentences by recently proposed encoder-decoder type of architectures which are mainly used for machine translation up to now. They obtained very impressive results and were able to beat state of art. This is mainly an application paper and the model in the paper is not novel.

The objectives of the paper are clearly stated with several empirical evidences supporting the authors' own claims. The fact that they were able to get better results with attention mechanism with even by just training on WSJ is quiet interesting, and that shows us the efficiency of using attention over regular encoder-decoder type architectures.

A few minor comments: * Authors mentioned that they used 3 LSTM layers in the model. But it was not very clear to me, whether if they used 3 layers in total (if yes how many layers did you used in encoder and how many layers did you have in the decoder) or both encoder and decoder had 3 layers. * It would be interesting to see some results on CCGBank as well. * I am curious about whether if a model trained with the objective of parsing a sentence can capture semantics as well. This might shed some interesting light to the syntax-semantics, or namely "Colorless green ideas sleep furiously" debate in generative semantics. You can evaluate the embeddings learned by this model.

Summary: This is a very well-written paper with vast amount of empirical experiments.

Submitted by Assigned_Reviewer_2

The paper tackles (constituent) syntactic parsing by mapping this prediction problem to a sequence-to-sequence alignment problem, and then essentially applying a method recently developed in the context of neural machine translation (LSTM-encoder-decoder with an attention mechanism). The resulting parsing model achieves state-of-the-art results when used in the standard supervised set-up (PTB WSJ) and improves further when estimated in a semi-supervised / co-training regime.

What I find especially interesting in this paper is that the attention mechanism is crucial for attaining good generalization properties: without using the attention mechanism LSTM achieves very poor results in the supervised setting. This is an interesting observation which may in principle generate future work focusing on refining the attention model (e.g., moving more in a direction of Neural Turing machines of Graves et al.).

This is also somewhat surprising that such simple linearization strategy led to state-of-the-art performance. This is another direction which may be interesting to explore.

Previous research into incremental parsing (psycho-linguistically inspired) may inform this work (e.g., predictive LR order). However, of course, it goes slightly against the declared agenda -- defining a general method ("keeping the framework task independent", l. 179, p.5).

The fact that LSTM is accurate (or at least not less accurate than LA PCFG) on long sentences is indeed surprising. Note though a related finding in Titov and Henderson (IWPT '07): they found that using recurrence (i.e. RNN-like modeling) resulted in substantial improvements specifically on longer dependencies when compared to a NN model which lacked recurrent connections (TF-NA). However, they also did not have a convincing explanation for this phenomenon.

Other relation with this previous work (as well as with SSN work of Henderson) seems quite interesting: in the submission the neural network essentially learns which previous hidden layers need to be connected to the current one. In this past work, alignments between hidden layers were defined by linguistic considerations (~ proximity in the partial syntactic tree).

So they essentially used a hand-crafted attention model.

I would appreciate a bit more analyses of the induced alignment model. So far only examples where the attention moves forwarded are considered; it would be interesting to see when jumps back are introduced (as at the very right in Figure 4). Also, it may be interesting to see the cases where the alignments were not peaked at a single input word (i.e. not a delta function).

Alignments in the NMT model of Bahdanau et al. seem a little less peaked; I am wondering why.

One difference I seem to notice between the attention-based NMT and the parsing model is that the alignments in the NMT work are computed based on hidden layers of RNNs running *both* in forward and backward direction. I am wondering if the authors could provide an intuition why they decided to limit themselves to only using a forward RNN. It may seem that using some kind of look-ahead should be useful.

Summary: The paper shows how a recently developed neural machine translation method can be applied to syntactic parsing.

The results are very competitive and the analysis are quite interesting.

Submitted by Assigned_Reviewer_3

Summary: This paper defines parsing as an efficient, domain-agnostic, linearized tree, sequence-to-sequence model. They achieve strong results when they add attention and also use large automatically-parsed corpora. Analysis and out-of-domain results are also presented.

Quality: The idea of a linearized tree, sequence-to-sequence model for constituency parsing is interesting and achieves high efficiency and accuracy. The ablation and analysis is useful. The paper is well-written but is missing certain explanations and definitions useful for a general audience. Also, there are certain issues with the experimental setup that needs clarifications or extra experiments:

-- when adding unlabeled parsed sentences from the Web, the authors should explain how much overlap there is with the dev or test sentences, because then it is more like LSTM memorizing parser-outputs during training and replicating during testing (which is then more like replicating output of a tri-trained parser). -- the comparison doesn't really seem too fair because one needs to see how a standard parser (say BerkParser) does on tons of training data (even if it is much slower, which is still a win for this work), gold or tri-trained data. The authors do train a BerkParser on more data for out-of-domain experiments so why not report that for Table1 too? Esp. because the BerkParser trained on more data does beat their work on out-of-domain. -- how are all the model decisions made, e.g., reversing, tag normalization, 510-dim for word vectors? On the dev set? Please clarify. -- why is there an in-house implementation of Berkeley-Parser when it is publicly available? What are the differences, qualitatively and quantitatively?

Other suggestions or comments: -- in Sec 3.2, explain 'the semi-supervised results' for a general audience. Similarly, explain, why the non-terminal labels are added as subscripts to the parentheses. -- more analysis is needed for the surprising result of not needing POS tags -- since reversing the input sentence helps, did you try the bidirectional encoding?

Clarity: The paper is mostly well-written and well-organized, though is missing certain explanations and definitions useful for a general audience.

Originality and significance: The paper is original and has a useful impact in terms of defining parsing as an efficient, domain-agnostic, linearized tree, sequence-to-sequence model. It achieves strong results and provides good analysis. Previous work such as linear-time incremental parsing and NN parsers are similar to this work but this work achieves higher accuracy and efficiency.
Summary: This paper defines parsing as an efficient, domain-agnostic, linearized tree, sequence-to-sequence model. They achieve strong results when they add attention and also use large automatically-parsed corpora. However, there are certain issues with the experimental setup that needs clarifications or extra experiments (e.g., overlap between unlabeled training data and eval sets, comparison to BerkParser trained on more gold or tri-trained data, model decisions on dev or test, reason for in-house reimplementation).

Submitted by Assigned_Reviewer_4

This paper proposes a method to learn a syntactic constituency parser using an attention-enhanced sequence-to-sequence model. The paper claims that it obtains state-of-the-art results on WSJ and performs comparatively to Berkeley parser.

The problem of syntactic parsing is an important one in NLP, and is a fundamental pre-processing step for more advanced tasks such as named entity extraction or relation extraction. Therefore, any improvement on syntactic parsing is likely to have a significant impact on the community. To this end the application of LSTM-based sequence-to-sequence coding is both timely and important.

Additional strong points of the proposed parser are its speed, and domain adaptability.

However, I have several concerns that I would wish the authors could further clarify as mentioned below.

The title of the paper and its relevance to the current work is somewhat unclear. I do understand that sequence-to-sequnce mapping has been previously applied for machine translation, but what exactly do you mean by "grammar as a foreign language"? The relevance of the title to the proposed method is never explicitly discussed in the paper. I guess you wanted to hint that you could "learn" the grammar of a language using sequence-to-sequence models. But the evaluation tasks are all monolingual adaptation task and not cross-lingual parsing tasks.

There are several statements in the paper that require further clarifications or analysis. For example, it is mentioned that replacing all pos tags by XX gave an F1 improvement of 1 point. Why is this? Removing POS info means you are loosing many important features that have been found to be useful for syntactic parsing.

What is the significance that the constructed parser matches Berkeley parser with 90.5 F1? Should not it be the case that the comparison should be done against an external test dataset and not the parser that generated the train data in the first place?

The degree of novelty w.r.t. machine learning or deep learning is minimal in the paper considering that it is using a previously proposed sequence-to-sequence encoding method based on LSTM for parsing. On the other hand, the claims of the paper and application task (parsing) is more relevant to the NLP community. The paper would be much appreciated and recognized at an NLP conference such as ACL or EMNLP than at NIPS. Therefore, it would be more suitable to submit this work at an NLP related conference.

The problem of domain adaptation of parsers has been considered in the NLP community and arguably the most famous benchmark for evaluating this would be SANCL-2012 "parsing the web" task (see Overview of the 2012 Shared Task on Parsing the Web by Petrov and McDonald 2012). Unfortunately, this line of work is not discussed nor evaluated against as a benchmark dataset in the current paper, which makes it difficult to justify the claims about domain adaptability of the proposed method.

The robustness of the parser is also a concern. It is mentioned that all 14 cases where the parser failed to produce a balanced tree are fragments or not ending with proper punctuations. Unfortunately, majority of the text found in social media such as twitter would be such cases. It is not discussed in the paper how you could address this issue (in particular that you claim the domain adaptability of the parser) in a more systematic manner than simply adding brackets to balance it.
Summary: This paper proposes a method to learn a syntactic constituency parser using an attention-enhanced sequence-to-sequence model. The paper claims that it obtains state-of-the-art results on WSJ and performs comparatively to Berkeley parser.

Submitted by Assigned_Reviewer_5

The paper looks a bit like an application; perhaps more focus should be given to novelty and analysis. As it is I feel that this would be a good workshop paper.

In particular, it is not quite clear what part of the model is novel and what is previous work. Given the previous work on attention with RNNs: is the attention part novel?

The title, 'grammar as a foreign language' seems confusing to me, since grammar is not a language, and I feel that the title doesn't describe the method well.

This lstm uses forget gates, correct? Then Gers et al (2000) should be cited, because forget gates were absent from the original lstm. I also think that Pollack used RAAM networks to learn small parse trees in the early 1990s - this should be cited as well.

Summary: Light review of Paper #1584: "Grammar as a Foreign Language"

This paper shows an interesting application of attention-based lstm to sentence parsing, using a dataset created by traditional parsers, achieving state-of-the-art results on certain datasets.

Author Feedback
Author rebuttal: We thank the reviewers for their helpful and extensive comments, which we address below. We write BP for BerkeleyParser.

Reviewer 1
We want to thank this reviewer for pointing out interesting related work that we were not familiar with and for emphasizing the main points of our work.

As suggested, we will look for more interesting examples on how the attention mechanism operates. We may also publish raw attention masks and our parsed results on the development set.

Regarding peakiness of attention, we see parsing as an almost deterministic function, whereas translation is more vague. Indeed, we attain a low perplexity (~1) compared to ~5 in translation models. So the parsing model is more confident, which could explain the peakiness in the attention.

Regarding bidirectionality, the model has, in principle, access to the full context because we provide hidden activations from the last timestep to the decoder. However, it could help even more, and we plan to investigate it.

Reviewer 2
We are grateful to this reviewer for making a very good point regarding overlap from sentences on extra data. We agree that having extra unsupervised data increases the amount of intersection. We did not investigate this in the current version of the paper. Given that we have only trained the model for 5 epochs on the 10M corpus, we doubt that intersecting sentences have a big effect. But for the final version we will re-train our model and make sure there are no duplicates between the training and evaluation sets.

Regarding the performance of BP with extra training data: it only increased BP's F1 score on the WSJ dev and test set insignificantly (by 0.1), but we will add this to Table 1. Note that even when trained on human-annotated data we achieve competitive results vs. BP. When training on more data, our model improves while BP seems to saturate.

As for the decision about model parameters: all of them were made by measuring performance on the development set. The test set has been evaluated fewer than 5 times over months of research.

Regarding our in-house reimplementation of BP, this is necessary in order to integrate it better with our infrastructure and to be able to run large-scale distributed experiments. Its implementation follows closely the original work of Petrov. The only deviation is that it uses a feature-rich emission model (as in "Feature-Rich Log-Linear Lexical Model for PCFG-LA Grammars", Huang & Harper, IJCNLP 2011).

Minor suggestions: we did not try bidirectional encoders (see the reply to Reviewer 1). As for the reason why removing POS-tags gave an F1 improvement, it is indeed very interesting and we will discuss it more extensively in the final version.

Reviewer 3
(1) Regarding the title, we agree that it is a bit obscure. We are considering to modify the title or to add extra clarification in text to show what we meant -- that one could map a sentence to its grammar using the same techniques as mapping a sentence to its translation.

(2) As for the removal of POS tags, see the reply to Reviewer 2.

(3) "What is the significance that the constructed parser matches BerkeleyParser (BP) with 90.5 F1?" We are sorry for this misunderstanding, we should have formulated it more carefully. By "match" we mean that the final F1 scores of BP and our model trained on the same data are similar -- we do not compare the output of our model with the output of BP. We, indeed, use an external test dataset with gold labels (the standard WSJ test set) and for the 90.5 score we trained only on the standard WSJ train set.

(4) Regarding novelty, we feel that the data-efficiency observation with regards to the attention mechanism and the fact that we can generate trees (it was conjectured that recursive Neural Nets are necessary) are significant machine learning contributions.

(5) The reviewer mentions the SANCL-2012 task and says "this line of work is not discussed nor evaluated". This looks like an oversight: we did use the SANCL-2012 WEB task [8] as discussed in line 290 (we followed the more recent recipe in Weiss et al., ACL'15). Further, we also use QTB [9] in addition to WEB [8] and report the performance of our model (which beats all numbers form [8]). Good performance on these tasks gives us reasons to claim domain adaptability and robustness. We will put more focus on it in the final version and we encourage the reviewer to re-read this part (Page 6, Performance on other datasets).

Reviewer 5
(1) Regarding layers in encoder / decoder: we used 3 layers on both encoder and decoder, we'll work to clarify this in the final version.

(2) We did visualize the embedding learned by this model (both word and sentence), but did not add this part due to space limitations. We may add it as an appendix in the final version.

Reviewer 6
Regarding novelty and the title, please see our response to Reviewer 3. As for the extra reference, we will add it in the final version.